# Effect of Steam on Carbonation of CaO in Ca-Looping

**DOI:** 10.3390/molecules28134910

**Published:** 2023-06-22

**Authors:** Ruzhan Bai, Na Li, Quansheng Liu, Shenna Chen, Qi Liu, Xing Zhou

**Affiliations:** 1College of Chemical Engineering, Inner Mongolia University of Technology, Huhhot 010051, China; 20211800161@imut.edu.cn (R.B.);; 2Hebei Key Laboratory of Inorganic Nanomaterials, School of Chemistry and Materials Science, Hebei Normal University, Shijiazhuang 050024, China; 3College of Chemistry and Chemical Engineering, Taiyuan University of Technology, Taiyuan 030024, China; 4College of Zhongran, Hebei Normal University, Shijiazhuang 050024, China

**Keywords:** Ca-looping, steam, carbonation conversion, catalysis

## Abstract

Ca-looping is an effective way to capture CO_2_ from coal-fired power plants. However, there are still issues that require further study. One of these issues is the effect of steam on the Ca-looping process. In this paper, a self-madethermogravimetric analyzer that can achieve rapid heating and cooling is used to measure the change of sample weight under constant temperature conditions. The parameters of the Ca-looping are studied in detail, including the addition of water vapor alone in the calcination or carbonation stage and the calcination/carbonation reaction temperatures for both calcination and carbonation stages with water vapor. Steam has a positive overall effect on CO_2_ capture in the Ca-looping process. When steam is present in both calcination and carbonation processes, it increases the decomposition rate of CaCO_3_ and enhances the subsequent carbonation conversion of CaO. However, when steam was present only in the calcination process, there was lower CaO carbonation conversion in the following carbonation process. In contrast, when steam was present in the carbonation stage, CO_2_ capture was improved. Sample characterizations after the reaction showed that although water vapor had a negative effect on the pore structure, adding water vapor increased the diffusion coefficient of CO_2_ and the carbonation conversion rate of CaO.

## 1. Introduction

It is believed that climate change is mainly caused by the increased CO_2_ concentration in the atmosphere [1]. Fossil fuel combustion systems, such as coal-fired power plants, are one of the major fixed sources of CO_2_ emissions [2,3]. Ca-looping is a high-temperature and low-cost technology that is under study for capturing CO_2_ from fossil fuel combustion [4,5].

Ca-looping systems use calcium carbonates that are typically derived from limestone or dolomite and are regenerable, abundant, and cheap sorbents [6,7]. 

Ca-looping is based on the reversible reaction described in Equation (1). The forward reaction is known as carbonation, and the reverse reaction is known as calcination. Calcination is an endothermic process which readily goes to completion under a wide range of conditions [8,9].


(1)
$$\mathrm{CaO}+{\mathrm{CO}}_{2}\overset{\mathrm{carbonation}}{\underset{\mathrm{calcination}}{⇄}}{\mathrm{CaCO}}_{3}$$


Ca-looping can be achieved using a double fluidized bed system [10,11]. The solid adsorbent is continuously circulated between two interconnected fluidized bed reactors. Carbon dioxide (CO_2_) in flue gas is absorbed by calcium oxide (CaO) in a carbonizer at about 650 °C. Calcium carbonate (CaCO_3_) is then transferred to the regeneration reactor (calciner) [12]. 

In the calcinator, calcium carbonate is decomposed into calcium oxide (CaO) and carbon dioxide at 900 °C. The regenerated CaO is returned to the carbonizer, leaving a pure carbon dioxide stream, resulting in simultaneous regeneration of the adsorbent (CaO → CaCO_3_ → CaO) [13]. Under actual conditions, the energy required to regenerate the adsorbent (calcined carbonate) is provided by firing coal or biomass using oxy-fuel technology to avoid dilution of the CO_2_ stream with N_2_ from air [14].

It has been widely reported [15,16] that the conversion ratio of CaO to CaCO_3_ decreases with an increase in calcination/carbonation cycles. This phenomenon is considered to be related to the decreased active surface of adsorbent caused by sintering [17]. It seems that sintering plays a principal role in the decreased reactivity of the Ca-based sorbents. Sintering reduces the surface area of the sorbents and the number of active sites [18,19].

There are many factors affecting the sintering of Ca-based sorbent, including reaction temperature, reaction duration, cycling numbers, and reaction atmosphere [20,21]. For the reaction atmosphere, water vapor plays an important role [22]. On the one hand, it has been reported that water vapor may accelerate the sintering of CaO [23]; on the other hand, water vapor can promote both reaction rate and conversion between CaO and CO_2_. Wang et al. [24] studied the effect that water vapor has on the carbonation of CaO in oxy-fuel CFB combustion conditions. Water vapor significantly improves absorption of carbon dioxide [25]. Donat et al. [26] investigated the influence that steam has on the reactivity of four kinds of limestone in an atmospheric pressure bench-scale bubbling fluidized bed (BFB) reactor that was made of a quartz tube and using periodically changing temperatures. In a word, water vapor is one of the factors that affect sintering in the Ca-looping process, and its role requires further study.

The effect of steam atmosphere on the Ca-looping calcination/carbonate process in laboratory tests still faces some challenges. A fluidized bed reactor is very good at simulating actual working conditions of Ca-looping cycles, but it is difficult to obtain reaction kinetics data and to evaluate the influences that individual operation parameters have. TGA is commonly used to measure calcination/carbonation kinetics [27,28,29]. Therefore, more researchers have used TGA to conduct Ca-looping experiments. However, the heating rate of TGA is usually set as 10–20 K/min, and diffusion resistance is serious, which is inconsistent with the factor that the Ca sorbent is cycling between the high-temperature calcination reactor (about 900 °C) and low-temperature carbonation reactor (about 650 °C) [30].

In this work, a custom-made thermogravimetric analyzer system with a large sample scope (100–300 mg) is used, which allows fast temperature and atmosphere switching when considering the water vapor atmosphere. On the one hand, it can simulate the sudden temperature change of the CaO adsorber in the calciner and carbonation reactor to the maximum extent; on the other hand, it can switch the temperature and atmosphere as quickly as 5–15 s and thus can exclude the long temperature/atmosphere switching time in the conventional TGA and the influence of the residence time on the sintering. In turn, the effect of water vapor in the calcination and/or carbonation stages, including conversion, kinetics, and surface morphology, can be studied more accurately. In the course of this paper, the temperature rise and fall rates of the samples are fully considered. In addition, the effect of CO_2_ diffusion in the N_2_ and N_2_-H_2_O mixture atmosphere on CO_2_ diffusion was calculated. The results of this work will help in producing more effective designs and control strategies of the Ca-looping process.

## 2. Results and Discussion

### 2.1. Effects of Steam during Both Calcination and Carbonation

A multicycle test of eight cycles was performed using BD limestone. Figure 1 shows the results that were obtained with two different experimental methods. As mentioned in the Introduction, with an increase in the cycle number, the amount of newly formed CaCO_3_ decreases and the time required for its decomposition decreases [14].

In Figure 1a, both calcination and carbonation time were fixed; the calcination period was 360 s and carbonation time was 300 s, whether calcination was completed or not. The calcination time in Figure 1b is adjusted with the increasing cycling number; once calcination was completed, the calcination was immediately terminated, and then, carbonation was carried out. On this basis, the influence of water vapor is also considered. Under the two different experimental methods, the carbonation conversion ratios were different. Generally, the conversion ratios in (a) were slightly lower than those in (b). This was obviously a result of sintering because of the excessive calcination time. Many studies used testing procedures that were similar to method (a). However, method (b) is considered more reasonable and was used in this study.

In addition, several characteristics are noted in Figure 1. Steam had a positive effect on CO_2_ capture in Ca-looping. Steam increased the decomposition rate of CaCO_3_ and enhanced the subsequent carbonation conversion of CaO. With 20% steam, both the reaction rate and calcium conversion ratio improved during all eight of the carbonation cycles compared to the results that were obtained without steam.

Figure 2 shows the conversion rate of carbonations in Figure 1b. Steam had a positive influence on carbonation, and this effect was more pronounced at higher steam concentrations and increasing cycle numbers.

### 2.2. Steam Addition during Calcination or Carbonation

In terms of the eight calcination/carbonation cycles experiments, further work is needed to determine the role of water vapor in each of the calcination and carbonation processes. Thus, a new test method is suggested here: steam is present in either calcination or carbonation but not in both reaction periods. In addition, the steam concentrations (0%, 10%, and 20%) are also considered. The carbonation conversions of CaO from BD limestone are shown in Figure 3.

There was little difference in carbonation conversion for the CaO that calcined with steam until the third cycle. For the first carbonation cycle, the conversion was almost the same (~0.63) for all three of the calcination conditions. The differences became obvious at higher cycle numbers. For example, in the eighth cycle, the conversion ratio was 0.23 for CaO that was calcined without steam, 0.21 with 10% steam, and 0.19 with 20% steam. It can be concluded from the results shown in Figure 3 that the presence of steam in the calcination stage led to lower carbonation conversion of CaO, and when there is more steam, the sintering is more severe, which is consistent with previous studies [31] in showing that steam can accelerate the sintering of CaO and result in decreased surface area and porosity.

The first, fourth, and eighth carbonation cycles were recorded to help understand the effect of steam addition on carbonation reaction, and the results are shown in Figure 4. For the first cycle, there was almost no difference between CaO that was calcined with 0% and 20% steam. However, after four cycles, the differences became significant, especially after 50 s. Obviously, the addition of water vapor in the calcination shows a negative effect on the subsequent carbonation.

The effect of adding water vapor exclusively during the carbonation process is shown in Figure 5. For all eight cycles, the presence of steam in the carbonation stage improved the CO_2_ capture capacity. The relative magnitude of the effect became more significant with an increase in the cycle number. For cycles 1 and 2, the differences in the carbonation conversion were minor at different steam concentrations. For the eighth cycle, the carbonation conversion was 0.33 with 20% steam, which is about 44% higher than the value of 0.23 without water vapor. Water vapor has a positive effect on the carbonation of CaO.

Figure 6 shows the carbonation of CaO versus time for cycles 1, 4, and 8 when water vapor was only added in the carbonation stage. The effects were obvious from the start to the end. In this work, a sample was quickly moved into the furnace for calcination when the set temperature was reached and maintained. The phenomenon that is observed in Figure 6 may be easily explained by the catalytic effect of steam in the fast chemical control stage, as proposed by Wang [24] and Yang [32].

### 2.3. Reaction Temperature

The reaction temperature is one of the most important parameters for Ca-looping [14,33]. Calcination at 900 °C and carbonation at 650 °C were used in Section 2.1 and Section 2.2. To further investigate the effects of temperature, more experiments were conducted. One set was performed at a carbonation temperature of 650 °C and calcination at 950 °C and 1000 °C. The second set of tests were conducted at a fixed calcination temperature of 900 °C and carbonation at 700 °C and 750 °C. The results are shown in Figure 6 and Figure 7.

Several observations are made from Figure 7. The first is that for all three of the calcination temperatures, the conversion of CaO was improved by the presence of steam. The levels of improvement were very similar. The second observation is that the CaO conversion was almost the same (about 0.61) regardless of whether it was with 20% steam or without at 900 °C and 950 °C. However, the CaO carbonation conversion ratio decreased sharply at 1000 °C, and the conversion was about 0.45. The carbonation conversion ratio of CaO that was calcined at 900 °C and 950 °C for the first cycle was about 61%, but it was only 45% for the CaO that was calcined at 1000 °C.

Although adding steam can compensate for some of the lost CO_2_ capacity, the carbonation conversion was still lower than that of the CaO that was calcined at the two lower temperatures even without added steam. A high calcination temperature led to sorbent reactivity decay in two ways: ion migration in sorbent crystal structure and sintering. Sintering decreased the sorbent surface area, and it is known that maximum conversions depend on the surface area of sorbents [34].

The initial calcination of BD limestone was compared in Figure 8. The calcination time at 900 °C was almost double that at 950 °C. It is known that the sintering of CaO is related both to temperature and to calcination time. Although higher temperature might have caused a higher rate of sintering, the shorter calcination time at 950 °C might reduce the overall sintering effect on CaO. This might be why the CaO calcined at 900 °C and 950 °C had similar CO_2_ capture ability, as shown in Figure 9.

Figure 9 showed carbonation conversions versus cycle number at 650, 700, and 750 °C for BD limestone. At 650 °C and 700 °C, 20% steam improved the carbonation conversion of CaO compared to the sample without steam, and it seems that 700 °C produces better conversion than 650 °C. However, when the carbonation temperature was increased to 750 °C, the carbonation conversion decreased greatly, even for the first cycle.

As also seen in Figure 9, steam improved the conversion of CaO to CaCO_3_ at all three carbonation temperatures. However, the magnitude of improvement was different, and the improvement was less pronounced at higher temperatures. A probable reason for this is that steam plays a catalytic role in the carbonation process. As a necessary process for a catalyst, steam must be adsorbed onto the surface of CaO. However, higher temperature had a negative effect on the adsorption process [18]. Hence, with an increase in the carbonation temperature, the catalytic effect of steam decreased.

To confirm this hypothesis, more carbonation experiments were conducted at different temperatures. Figure 10, which shows carbonation conversion versus time curves of CaO at four temperatures, clearly shows that steam had an obvious effect on the CaO carbonation rate during the initial fast reaction stage but not during the following slow diffusion-controlled reaction stage. In addition, steam has a greater catalytic effect on the carbonation reaction at lower temperatures. Linden et al. [35] found that a partial pressure of steam had a positive effect on the conversion of CaO to CaCO_3_ in the temperature range of 400~550 °C. Figure 10 also shows that the reaction was quite slow in the temperature range of 550~600 °C, although CaO achieved a reasonable conversion ratio over much longer time periods. Given that the volume of CO_2_-containing flue gas from coal-fired power plants is very large, a slower reaction rate to achieve high CO_2_ capture efficiencies requires the construction of very bulky adsorber units to considerably extend the contact time of CaO and CO_2_. Thus, from the results in Figure 10, it is determined that 650~700 °C is the most suitable temperature for carbonation for BD limestone.

### 2.4. Pore Structure of CaO in Looping and Gas Diffusion Coefficient

In Section 2.1–2.3, steam has important influences on the carbonation ability of CaO. The surface morphology [36] and pore characteristics [37] of CaO are important factors that affect its carbonation in the looping cycle. Hence, comparing the surface morphology and pore characteristics of CaO that was calcined with different concentrations of steam may be helpful to understand the influence of steam. Figure 11 shows SEM images of the samples under each condition. In Figure 11a, CaCO_3_ is dense and nonporous. After calcination, the CaO shown in Figure 11b has a rich pore structure. After calcination in an atmosphere containing water vapor, in Figure 11c, the particle size of CaO increased, and some small particles grew. After carbonization in an atmosphere with water vapor addition, the surface of CaCO_3_ (in Figure 11d) is more compact, which corresponds to higher carbonation conversion.

The pore structures of CaO samples that were calcined in typical conditions were tested using N_2_ absorption, and the results are shown in Table 1.

Table 1 shows that regardless of whether or not there is water vapor, the specific surface area and pore volume of CaO decrease with increasing cycles. For example, without water vapor, the specific surface area of CaO decreased from 8.38 to 3.08 m^2^/g after the eighth cycle. Another observation is that there is very little difference in the CaO pore structure with or without vapor. However, in Section 2, it was clear that water vapor has a positive effect on CaO carbonation. Hence, the fact that the pore structures were similar with or without water vapor may indicate that the presence of steam predominantly plays a catalytic role in this case.

It was shown that steam did not have a significant influence on the preference of a Ca-based sorbent to adsorb CO_2_; thus, more explanations are needed to account for this observation. Mass transfer is a key step in these reactions, and Figure 12 shows the effective diffusion coefficient of CO_2_ (D*e*) in a N_2_ and N_2_ + H_2_O mixture. The D*e* of CO_2_ also increased with temperature and steam addition. It is clear that adding steam could improve the diffusion of CO_2_ from the gas phase to the solid sorbent.

On the basis of the surface morphology, pore structure, and D*e* of CO_2_, a hypothesis is suggested here that there are three ways for steam to have an effect on the calcination and carbonation of a sorbent in looping. These three are high-temperature sintering, water vapor increasing CO_2_ diffusion ability, and the catalytic effect of H_2_O. The sintering of CaO that was accelerated by steam during calcination decreases its carbonation ability, and this negative effect should decrease with more cycles. The catalytic effect of steam has little relationship with its pore structure. Thus, the carbonation ability of CaO that undergoes severe sintering should be almost the same regardless of whether the sintering is with steam or not.

## 3. Experiment

### 3.1. Sample Preparation

In this study, natural limestone (BD) was used as a raw material. After crushing and screening, BD with a particle size of 150~250 μm was selected. The samples were dried at 105 °C for 4 h. Analysis results of BD limestone are given in Table 2.

### 3.2. Experimental Parameters

The limestone samples underwent calcination and carbonation in two separate tube furnaces, and the sample weight was measured continuously using a detection system. The experimental setup is shown in Figure 13.

First, two tube furnaces were heated to the test temperature, with a reaction gas flow rate of 1200 mL/min. At 1200 mL/min, mass transfer is not the limiting factor of the reaction, based on a previous experiment.

Second, a limestone sample (about 300 mg) was loaded into a quartz boat (110 mm long, 15 mm wide, and 12 mm deep).

Third, when the furnaces reached the test temperature and stabilized, the limestone sample was quickly moved into the calciner. When the weight loss stopped (signaling the end of the calcination period), the sample was quickly moved into the carbonation furnace to carbonate with the synthetic flue gas. After carbonation, the sample was returned to the calcination furnace.

Finally, the above procedure was repeated for 8 complete calcination/carbonation cycles. The time needed to move the sample holder between the furnaces was less than 2 s.

The steam was added through a steam-generating unit, which has a programmable injection pump that feeds water into the evaporator. The temperature of the evaporator was maintained at about 220 °C. As a result, when the liquid water enters the evaporator, it is immediately converted into steam.

The CO_2_ uptake capacity of the samples is calculated from mass changes using Equation (2):(2)$$X_{N}^{}=\frac{m_{carb}^{N}-m_{cal}^{N}}{{\beta m}_{0}^{}}\frac{M_{\mathrm{CaO}}^{}}{M_{{\mathrm{CO}}_{2}}^{}}$$
where *X_N_* is the carbonation conversion of the sample, *N* is the cycle number, *m*_0_ is the initial mass (g) of the sample, $m_{carb}^{N}$ is the mass (g) of the recarbonated sample after the *N*th cycle, $m_{cal}^{N}$ is the mass (g) of the calcined sample after the *N*th cycle, and *β* is the content of CaO in the initial sample. *M*_CaO_ and $M_{{\mathrm{CO}}_{2}}^{}$ are the molar weights (g/mol) of CaO and CO_2_, respectively.

Diffusivity of CO_2_ (D*e*) was then calculated using the following formulas [38]:

D is the intrinsic diffusion coefficient:(3)$$D_{}^{}=\frac{1}{1/D_{{\mathrm{CO}}_{2},N_{2}-H_{2}O}^{}+1/D_{K}^{}}$$

$D_{{\mathrm{CO}}_{2},N_{2}-H_{2}O}^{}$ is the effective diffusion coefficient of CO_2_ in the mixture of N_2_ and H_2_O:(4)$$D_{{\mathrm{CO}}_{2},N_{2}-H_{2}O}^{}=\left(1-y_{{\mathrm{CO}}_{2},N_{2}-H_{2}O}^{}\right)/\sum_{i=2}^{k}\left(y_{i}^{}/D_{1i}^{}\right)$$

D*_K_* is the Knudsen diffusion coefficient:(5)$$D_{K}^{}=97r_{K}^{}\sqrt{\frac{T_{}^{}}{M_{{\mathrm{CO}}_{2}}^{}}}$$

D*e* is the effective diffusion coefficient, which is primarily controlled by Knudsen diffusion:(6)$$D_{e}^{}=D_{}^{}\frac{\varepsilon_{p}^{}}{\tau }$$
(7)$$\varepsilon_{p}^{}=V_{p}^{}\rho_{p}^{}$$
where *ε_i_* is the porosity of the particle, *V_p_* is the quality of the catalyst pore volume (kg/m^3^) for the unit, *ρ_p_* is the density (kg/m^3^) of the sorbent particles, and *τ* is the labyrinth factor, which has a value of 3 here.

## 4. Conclusions

In this paper, the effects of water vapor on the calcination of calcium-based sorbents and the carbonation of calcium-based carbon dioxide sorbents were studied using a self-made thermogravimetric analyzer that can achieve rapid cooling. The results show that when water vapor is present in both the calcination and carbonation periods, it has an overall positive effect on CO_2_ capture in the Ca-looping process. Water vapor increases the calcination rate of CaCO_3_ and enhances the subsequent carbonation conversion of calcined CaO. However, the presence of water vapor in the calcination of limestone only leads to the deactivation of CaO, and this can result in lower carbonation conversion of CaO. The presence of water vapor in the carbonation stage improved CO_2_ capture for all eight of the cycles tested. There are only insignificant differences in the carbonation conversion ratio for CaO samples that were calcined at 900 °C and 950 °C. This is likely caused by the combined effect of the calcination rate and the sintering time. It seems that carbonation at 700 °C has a better conversion ratio than that at 650 °C, and the carbonation conversion was much worse at 750 °C. Water vapor accelerates the sintering of CaO during calcination. On the other hand, the effective diffusion coefficient D*e* of CO_2_ increases with the addition of water vapor and an increase in temperature.

## Figures and Tables

**Figure 1 molecules-28-04910-f001:**
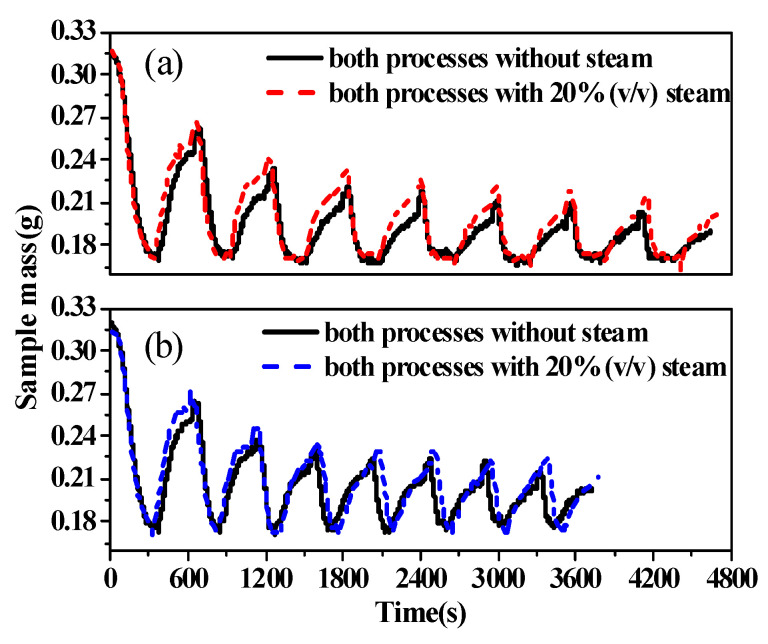
Sample weight vs. time for eight cycles of calcination/carbonation. (**a**) All calcination and carbonation periods lasted 300 s except the first calcination period, which was 360 s. (**b**) Calcination was completed and sent immediately to the carbonation furnace for carbonation, and all the carbonation periods were 300 s.

**Figure 2 molecules-28-04910-f002:**
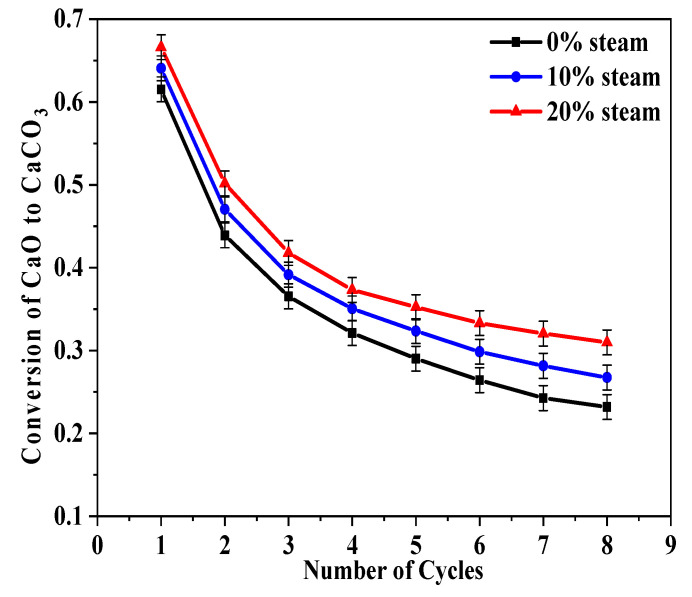
Influence of steam when it was present during both the calcination and carbonation stages on sorbent reactivity of BD limestone: calcination (80% CO_2_ + 0%/10%/20% H_2_O + O_2_ balance, 900 °C); carbonation (15% CO_2_+ 0%/10%/20% H_2_O + N_2_ balance, 650 °C).

**Figure 3 molecules-28-04910-f003:**
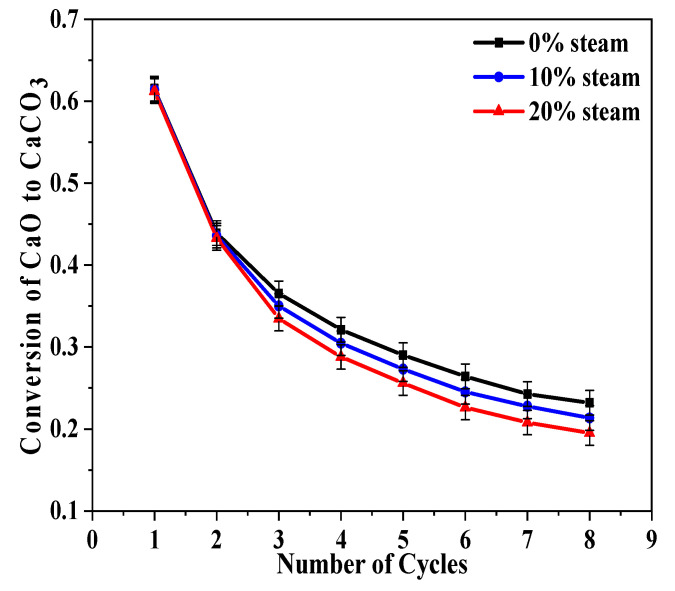
Influences of steam on conversion of CaO calcined from BD limestone: calcination (80% CO_2_ + 0%/10%/20% H_2_O + O_2_ balance, 900 °C); carbonation (15% CO_2_ + N_2_ balance, 650 °C).

**Figure 4 molecules-28-04910-f004:**
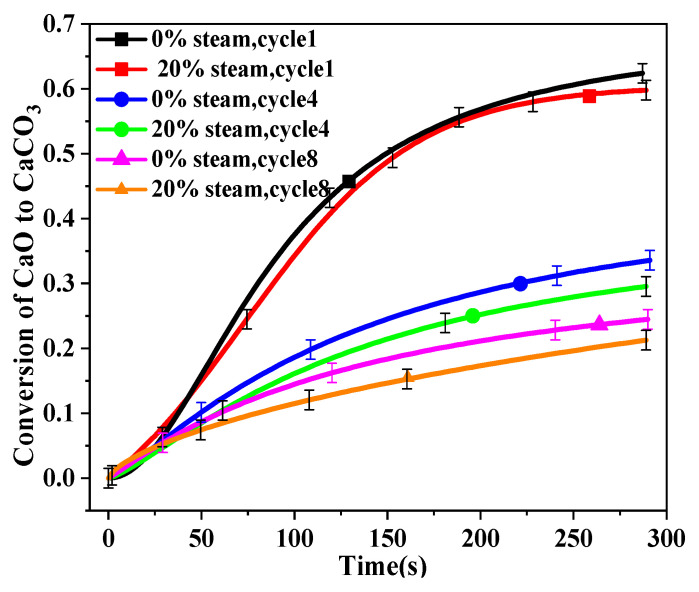
Conversion vs. time curves of BD limestone: calcination (80% CO_2_ + 0%/20% H_2_O + O_2_ balance, 900 °C); carbonation (15% CO_2_ balance N_2_, 650 °C).

**Figure 5 molecules-28-04910-f005:**
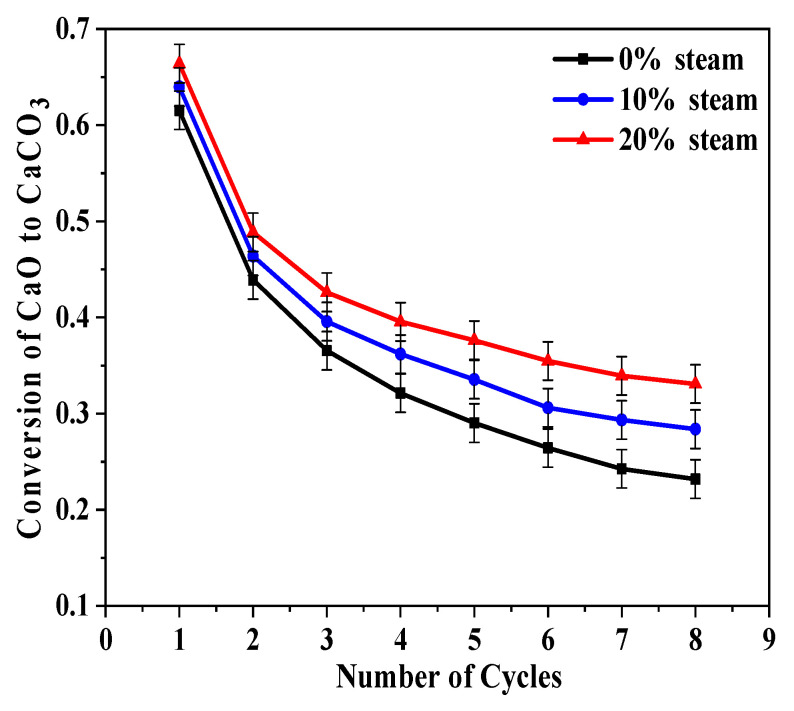
Influences that adding steam only during the carbonation stage has on sorbent reactivity of BD limestone: calcination (80% CO_2_, O_2_ balance, 900 °C); carbonation (15% CO_2_ + 0%/10%/20% H_2_O + N_2_ balance, 650 °C).

**Figure 6 molecules-28-04910-f006:**
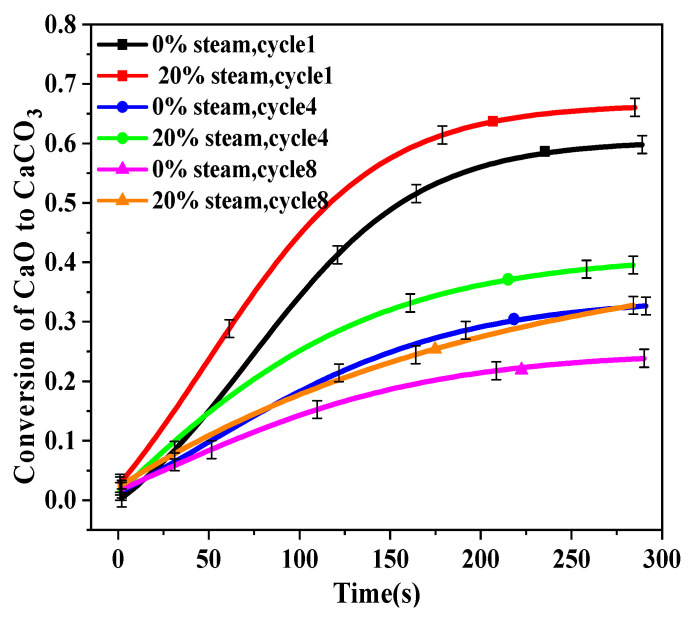
Conversion vs. time curves of different cycle numbers of BD limestone: calcination without water vapor (80% CO_2_ + O_2_ balance, 900 °C); only carbonation with vapor (15% CO_2_ + 0%/20% H_2_O + N_2_ balance, 650 °C).

**Figure 7 molecules-28-04910-f007:**
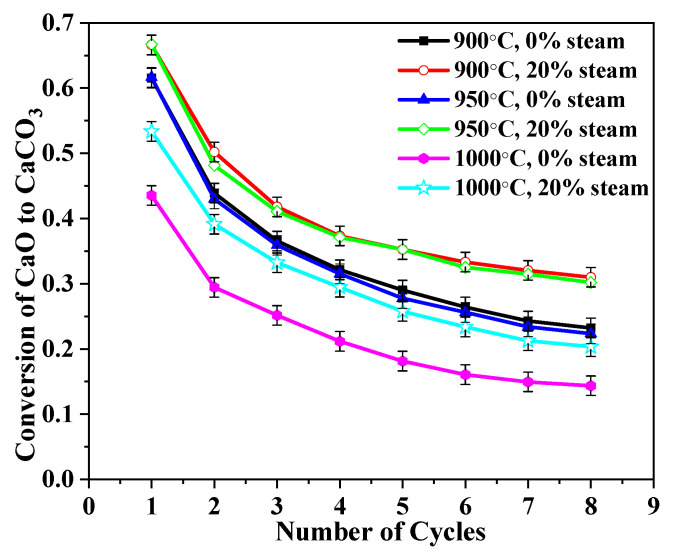
Influences of calcination temperature on sorbent reactivity of BD limestone: calcination (80% CO_2_ + 0%/20% H_2_O + O_2_ balance) at different temperatures; carbonation (15% CO_2_ + 0%/20% H_2_O + N_2_ balance) at 650 °C.

**Figure 8 molecules-28-04910-f008:**
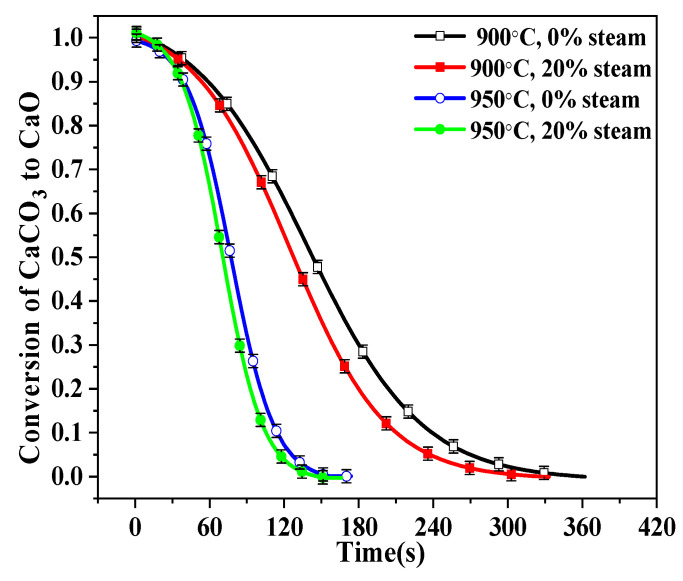
Decomposition curves of BD limestone calcination at 900 °C and 950 °C in 80% CO_2_ + 0%/20% H_2_O + O_2_ balance.

**Figure 9 molecules-28-04910-f009:**
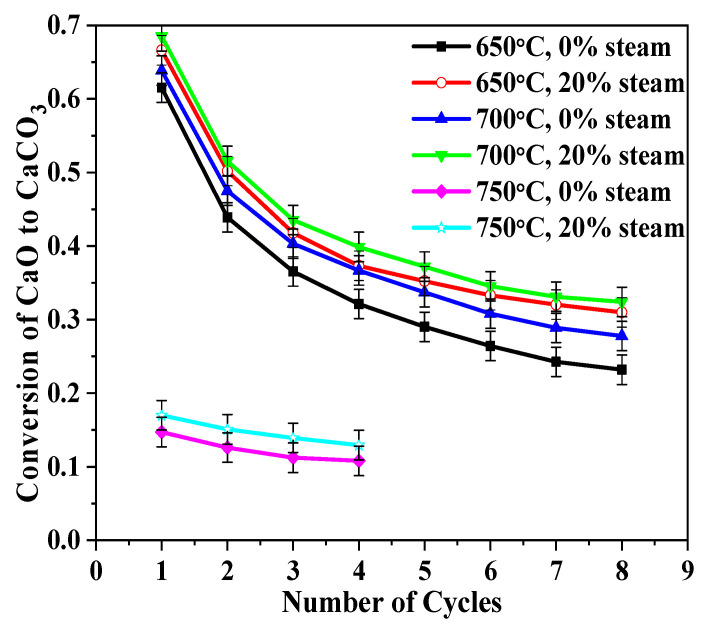
Influences of carbonation temperature on sorbent reactivity of BD limestone: calcination (900 °C, 80% CO_2_ + 0%/20% H_2_O + O_2_ balance); carbonation (15% CO_2_ + 0%/20% H_2_O + N_2_ balance).

**Figure 10 molecules-28-04910-f010:**
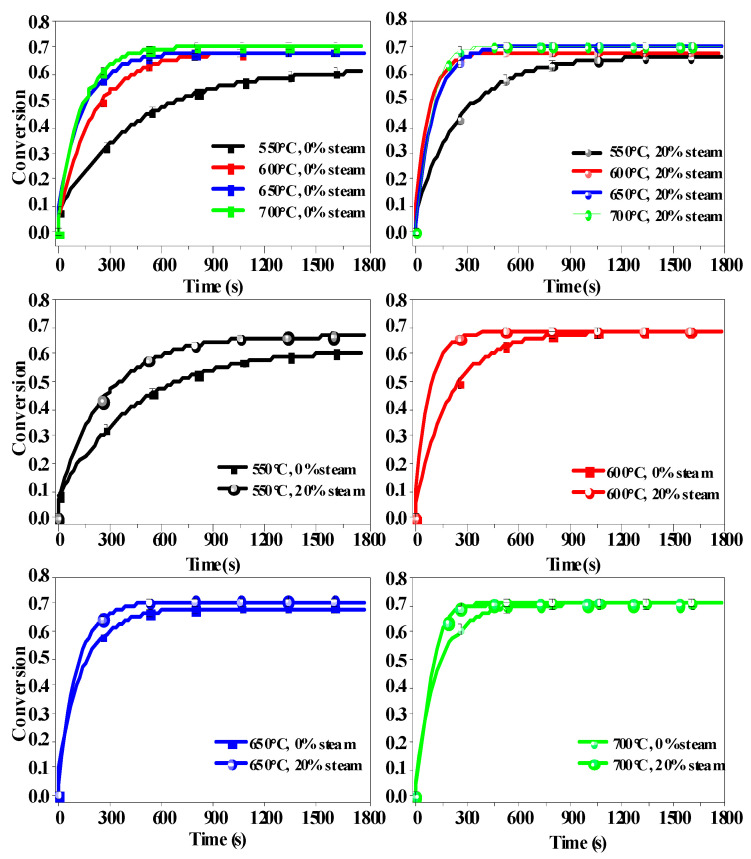
CaO carbonation conversion vs. time at different temperatures. (CaO was derived from BD limestone, calcination was at 900 °C in 80% CO_2_ + O_2_ balance, and carbonation was in 15% CO_2_ + 0%/20% H_2_O +N_2_ balance.)

**Figure 11 molecules-28-04910-f011:**
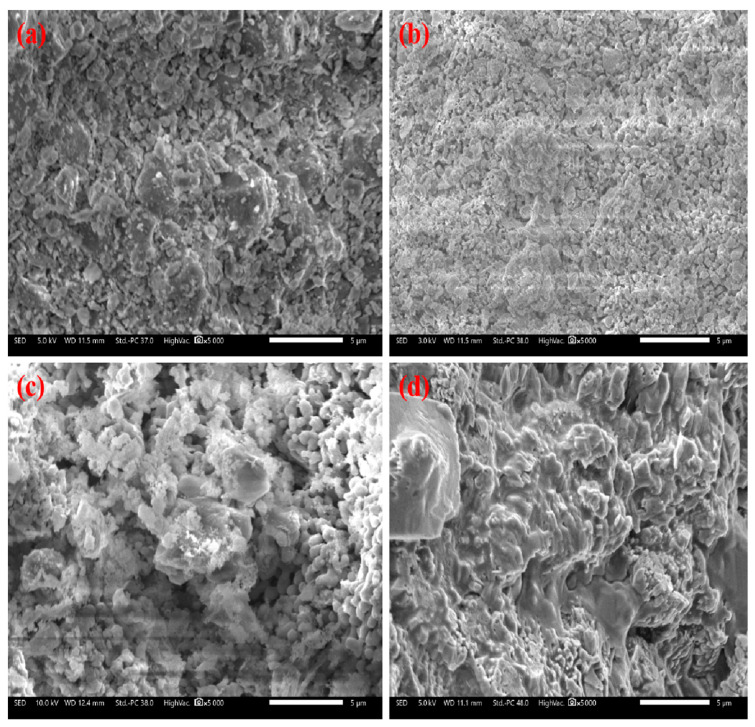
SEM images of samples under different conditions: (**a**) CaCO_3_, (**b**) CaO, (**c**) CaO (calcined with water), and (**d**) CaCO_3_ (calcined in water).

**Figure 12 molecules-28-04910-f012:**
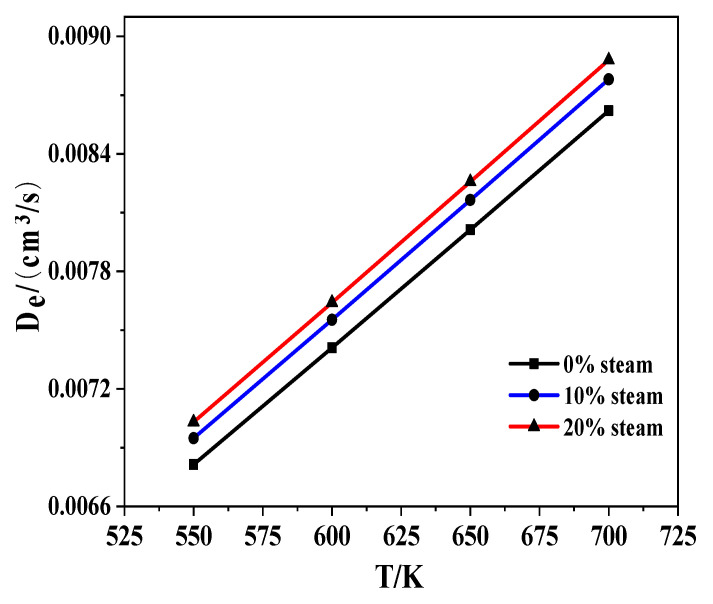
D*e* of CO_2_ vs. temperature in the presence of steam.

**Figure 13 molecules-28-04910-f013:**
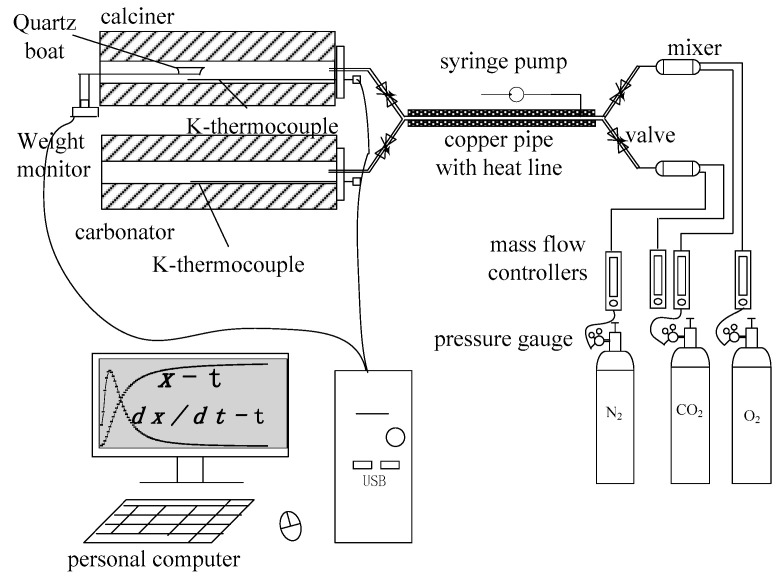
Tube furnace experimental system.

**Table 1 molecules-28-04910-t001:** Specific surface area and pore volume of CaO after first, third, and eighth calcinations.

	Surface Area m^2^/g			Pore Volume mm^3^/g		
	1st	3rd	8th	1st	3rd	8th
$\phi_{H_{2}O,\mathrm{cal}}$ = $\phi_{H_{2}O,\mathrm{car}}$ = 0%	8.38	4.45	3.08	28.28	11.45	5.65
$\phi_{H_{2}O,\mathrm{cal}}$ = $\phi_{H_{2}O,\mathrm{car}}$ = 20%	7.86	4.24	3.07	24.12	11.13	5.32

**Table 2 molecules-28-04910-t002:** Composition of limestone (wt.%).

Compound	SiO_2_	Al_2_O_3_	Fe_2_O_3_	TiO_2_	P_2_O_5_	CaO	MgO	SO_3_	Na_2_O	K_2_O	LOI
BD (wt%)	2.13	1.31	<0.55	<0.03	<0.03	53.22	1.48	<0.10	<0.20	0.13	41.02

## Data Availability

The data presented in this study are available on request from the corresponding author.

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
