# Peer review of "Effect of Steam on Carbonation of CaO in Ca-Looping"

_molecules, 2023, doi:10.3390/molecules28134910_

Round 1
Reviewer 1 Report (Previous Reviewer 3)
Despite the reforms, major revision still is required to boost the quality of the paper. The following comments should be considered before publishing this manuscript.
1- The following paragraph, page 2 and lines 46-49, “Ca-looping is based on the reversible reaction described in Eq.(1). The forward reaction in Eq.(1) is known as carbonation, and the reverse is known as calcination. Calcination is an endothermic process that readily goes to completion under a wide range of conditions” must be validated with the following refs: (1) Journal of CO2 Utilization, Volume 53, November 2021, 101747, and (2) Journal of Cleaner Production, Volume 384, 15 January 2023, 135579.
2- The 5th paragraph of the introduction, page 2 and lines 56-60, “In the calcinator, the CaCO3 decomposes into calcium oxide (CaO) and CO2 at 900℃, as the results, the adsorbent is simultaneously regenerated(CaO→CaCO3→CaO). The energy that is required to regenerate the sorbent (calcination of the carbonate) is provided by firing coal or biomass using oxy-fuel technology to avoid dilution of the CO2 stream with N2 from air.” needs to be validated with the following refs: (1) Process Safety and Environmental Protection Volume 144, December 2020, Pages 349-365, (2) Energy Conversion and Management, Volume 274, 15 December 2022, 116461.
3- Please edit the following typo errors: (1) Page 2, line 65, “sorbet”, (2) Page 5, line 165, “incereasing”. Also, please double-check the manuscript to avoid repeating grammatical and typo errors.
4- In the manuscript, “absorbent” and “absorption” have been written several times, but they must be replaced with adsorbent and adsorption.
5- Page 2, lines 78-82, please validate with up-to-date and pertinent refs.
6- Figure 1 is dispensable.
7- Page 5, line 172, Clarify Figure 2 a and 2 b, separately, in the title of Figure 2 to facilitate reading the manuscript.
8- The outcomes of Figure 2 have not been completely discussed, in detail. Mention the effect of calcination time on the water vapor impact. With terminating the calcination after completing the process, how much did the capture capacity change? Need to be brought up.
9- How did you conclude that the steam expedited the decomposition rate of CaCO3? Based on the TGA profile, the steam solely enhanced the CaO conversion.
10- As shown in Figure 3, With raising the steam content, the differences between sorption capacities are increasing as the number of cycles increases. How can you explain that?
11- As derived from Figure 6, with the cycle number increased, the difference between the sorption capacity of BD limestone under 0, 10, and 20% water vapor increases. The leading causes for this phenomenon should be noted in the manuscript.
12- Please support the following sentences with more refs that is proposed next to them.
“Fossil fuel combustion systems, such as coal-fired power plants are one of the major fixed sources of CO2 emissions.”: (1) Separation and Purification Technology, Volume 306, Part B, 15 February 2023, 122625. (2) DOI: 10.1007/s11705-022-2159-x.
“Ca-looping systems use calcium carbonates that are typically derived from limestone or dolomite, which sorbents are regenerable, abundant and cheap.”: (1) Journal of CO2 Utilization, Volume 65, November 2022, 102207. (2) Chemical Engineering Journal, Volume 417, 1 August 2021, 129194
13- Page 9, lines 272-273, The following sentence has been noted: “To examine the lack of difference in carbonation conversion for CaO that was calcined at 1000 ℃ and 950 ℃,” whereas, there is the significant variation in their cyclic conversion. Please revise this sentence.
More advanced structures for writing can be used in the manuscript. Overall, the manuscript can be considered for publication with this quality.
Author Response
1-The following paragraph, page 2 and lines 46-49, “Ca-looping is based on the reversible reaction described in Eq.(1). The forward reaction in Eq.(1) is known as carbonation, and the reverse is known as calcination. Calcination is an endothermic process that readily goes to completion under a wide range of conditions” must be validated with the following refs: (1) Journal of CO2 Utilization, Volume 53, November 2021, 101747, and (2) Journal of Cleaner Production, Volume 384, 15 January 2023, 135579.
Response:
Thank you for kind reminder. Litterateurs of (1) Journal of CO2 Utilization, Volume 53, November 2021, 101747, and (2) Journal of Cleaner Production, Volume 384, 15 January 2023, 135579 showed that the calcination of limestone is easy goes to completion under a wide range of conditions. The above litterateurs have been added in the revised manuscript.
[8] Heidari M, Tahmasebpoor M, Mousavi SB, Pevida C. CO2 capture activity of a novel CaO adsorbent stabilized with (ZrO2+Al2O3+CeO2)-based additive under mild and realistic calcium looping conditions.Journal of CO2 Utilization, 2021, 53: 101747.
[9] Mousavi SB, Heidari M, Rahmani F, Sene RA, Clough PT, Ozmen S. Highly robust ZrO2-stabilized CaO nanoadsorbent prepared via a facile one-pot MWCNT-template method for CO2 capture under realistic calcium looping conditions. Journal of Cleaner Production, 2023, 384: 135579.
Page 2, lines 46-49.
2- The 5thparagraph of the introduction, page 2 and lines 56-60, “In the calcinator, the CaCO3 decomposes into calcium oxide (CaO) and CO2 at 900℃, as the results, the adsorbent is simultaneously regenerated(CaO→CaCO3→CaO). The energy that is required to regenerate the sorbent (calcination of the carbonate) is provided by firing coal or biomass using oxy-fuel technology to avoid dilution of the CO2 stream with N2 from air.” needs to be validated with the following refs: (1) Process Safety and Environmental Protection Volume 144, December 2020, Pages 349-365, (2) Energy Conversion and Management, Volume 274, 15 December 2022, 116461.
Response:
Thank you for kind reminder. The reference of relevant literature is necessary to improve the accuracy of description, and according to the reviewer's opinion, relevant literature has been quoted.he above litterateurs have been added in page 2 and lines 56-60.
[13] Heidari M, Tahmasebpoor M, Antzaras A, Lemonidou AA. CO2 capture and flfluidity performance of CaO-based sorbents: Effect of Zr, Al and Ce additives in tri-, bi- and mono-metallic confifigurations. Process Safety and Environmental Protection, 2020, 14: 349–365.
[14] Heidari M, Mousavi SB, Rahmani F, Clough PT, Ozmen S. The novel Carbon Nanotube-assisted development of highly porous CaZrO3-CaO xerogel with boosted sorption activity towards high-temperature cyclic CO2 capture. Energy Conversion and Management , 2022, 274: 116461.
Page 2, lines 56-60.
3- Please edit the following typo errors: (1) Page 2, line 65, “sorbet”, (2) Page 5, line 165, “incereasing”. Also, please double-check the manuscript to avoid repeating grammatical and typo errors.
Response:
Thank you for kind reminder. hank you for kind reminder, and we are very sorry for this silly mistakes. The spelling errors have been corrected. And the manuscript have been double-checked . The contents are as follows: Page 2, line 65, “sorbents”; Page 5, line 165, “increasing”; Page 3, line 96, “adsorber”; Page 11,line 308,”adsorber”.
4- In the manuscript, “absorbent” and “absorption” have been written several times, but they must be replaced with adsorbent and adsorption.
Response:
Thanks for the reviewer's tolerance. We are very sorry for such low-level vulgar mistake. The “absorbent” and “absorption” have been replaced with adsorbent and adsorption in the revised manuscript.
The contents are as follows: Page 2, line 65, “sorbents”; Page 3, line 96, “adsorber”; Page 11, line 308,”adsorber”.
5- Page 2, lines 78-82, please validate with up-to-date and pertinent refs.
Response:
We are very sorry that the use of up-to-date is not appropriate. The up-to-date in the revised manuscript have been delete and some new references have been updated.
6- Figure 1 is dispensable.
Response:
I am sorry to say that figure 1 is not dispensable. Traditional Ca-looping is performed using thermogravimetric analysis systems or fixed bed reactor.
In the thermogravimetric analyzer, although the reaction temperature can be changed quickly, the reaction atmosphere is difficult to change quickly due to the small amount of samples and the large dead volume of the reactor. The switch between carbonation and calcination is mainly achieved by changing the reaction temperature. In the fixed bed reactor, the temperature is difficult to rise or fall rapidly due to thermal inertia, so the atmosphere is usually to realize the switch of calcination and carbonation.
The above situation becomes even more complicated when water vapor is considered. The equipment in this paper can realize the rapid change of both atmosphere and temperature in the case of adding water vapor. Figure 1 illustrates the innovation of the equipment in this paper, therefore the Figure 1 is not optional.
7- Page 5, line 172, Clarify Figure 2 a and 2 b, separately, in the title of Figure 2 to facilitate reading the manuscript.
Response:
Thank you very much for your careful reading of this manuscript. Detailed descriptions of figure 2 (a) and figure 2 (b) have been added. Figure 2. Sample weight vs. time for 8 cycles of calcination/carbonation. (a) all calcination and carbonation periods lasted 300 s except the 1st calcination period which was 360 s. (b)calcination was completed and sent immediately to the carbonation furnace for carbonation, and all the carbonation periods were 300 s.
please see the revised manuscript for details.
Page 5,lines179-182.
8- The outcomes of Figure 2 have not been completely discussed, in detail. Mention the effect of calcination time on the water vapor impact. With terminating the calcination after completing the process, how much did the capture capacity change? Need to be brought up.
Response:
Thanks for the reviewer's comments. As mentioned in issue 6, the device in this manuscript can realize the rapid change of both atmosphere and temperature in calcination and carbonation stage. Another feature of the device is that it can observe the calcination time online to avoid the decay of cyclic activity caused by too prolong the calcination time artificially.
Figure 2 compares the influences of fixed calcination time and dynamic calcination time on the reactivity of BD limestone, respectively. The detailed carbonation conversion results of dynamic experiment in figure 2 are shown in figure 3.
All experiments in this manuscript are mostly conducted on the basis of dynamic calcination experiments. Therefore, in order to avoid misleading readers, detailed results of fixed calcination in figure 2(a) are not shown.
9- How did you conclude that the steam expedited the decomposition rate of CaCO3? Based on the TGA profile, the steam solely enhanced the CaO conversion.
Response:
Thanks for the reviewer 's question. According to figure 9, the decomposition curve of CaCO3 at different temperatures determines the conclusion that water vapor promotes the decomposition of CaCO3.
In Fig.4-Fig.7, we can conclude that the addition of water vapor in the calcination stage will reduce the carbonation conversion rate, and the addition of water vapor in the carbonation stage will increase the carbonation conversion rate. On this basis, Figure 9 and Figure 13 show in detail the carbonation stage and calcination stage, water vapor on CaO carbonation, CaCO3 decomposition kinetics of detailed information.Thank the reviewer again for his valuable comments.
10- As shown in Figure 3, With raising the steam content, the differences between sorption capacities are increasing as the number of cycles increases. How can you explain that?
Response:
According to the literature[14][18]and experimental studies, the pore structure of Ca-based absorbers will change with the calcination/carbonation cyclicing number , which is the reason for the decrease the performance of Ca-based absorbers.
The addition of water vapor in the carbonation stage will increase the diffusion coefficient of CO2 to CaO. It is obvirous that the diffusion of CO2 is more important when the pore structure changes. As a result, in Figure 3, the differences between sorption capacities are increasing as the number of cycles increases. Thank you again for your careful reading of this manuscript.
[14] Heidari M, Mousavi SB, Rahmani F, Clough PT, Ozmen S. The novel Carbon Nanotube-assisted development of highly porous CaZrO3-CaO xerogel with boosted sorption activity towards high-temperature cyclic CO2 capture. Energy Conversion and Management, 2022, 274: 116461.
[18] Wang NN, Feng YC and Guo X. Atomistic mechanisms study of the carbonation reaction of CaO for high temperature CO2 capture. Applied Surface Science, 2020, 532: 147425.
11-As derived from Figure 6, with the cycle number increased, the difference between the sorption capacity of BD limestone under 0, 10, and 20% water vapor increases. The leading causes for this phenomenon should be noted in the manuscript.
Response:
Thanks for your careful reading our manuscript. The carbonation conversion gap between BD limestone with or without water vapor addition gradually increases with cycling number has been described in detail in the original manuscript. In addition, the description have been highlighted in the revised manuscript.
Page 7, lines 223-232.
12- Please support the following sentences with more refs that is proposed next to them.
“Fossil fuel combustion systems, such as coal-fired power plants are one of the major fixed sources of CO2 emissions.”: (1) Separation and Purification Technology, Volume 306, Part B, 15 February 2023, 122625. (2) DOI: 10.1007/s11705-022-2159-x.
“Ca-looping systems use calcium carbonates that are typically derived from limestone or dolomite, which sorbents are regenerable, abundant and cheap.”: (1) Journal of CO2 Utilization, Volume 65, November 2022, 102207. (2) Chemical Engineering Journal, Volume 417, 1 August 2021, 129194
Response:
Thanks for the valueable comments. The reference mentioned above by reviewer has been added in the revised manuscript and reference.
Page 2,lines 43-44 and lines 47-48.
References:
[2] Norbarzad MJ, Tahmasebpoor M, Heidari M, Pevida C. Theoretical and experimental study on the fluidity performance of hard-to-fluidize carbon nanotubes-based CO2 capture sorbents. Frontiers of Chemical Science and Engineering, 2022, 16(10): 1460-1475.
[3] Troya JA, Jiménez PSE, Perejón A, Moreno V, Valverde JM, Maqueda PAP. Kinetics and cyclability of limestone (CaCO3) in presence of steam during calcination in the CaL scheme for thermochemical energy storage. Chemical Engineering Journal, 2021, 417: 129194.
[6] Imani M, TPeridas G, Mordick Schmidt B. Improvement in cyclic CO2 capture performance and fluidization behavior of eggshell-derived CaCO3 particles modified with acetic acid used in calcium looping process. Journal of CO2 Utilization, 2022, 65: 102207.
13- Page 9, lines 272-273, The following sentence has been noted: “To examine the lack of difference in carbonation conversion for CaO that was calcined at 1000 ℃ and 950 ℃,” whereas, there is the significant variation in their cyclic conversion. Please revise this sentence.
Response:
Thanks for the valuable comments. The sentence: “To examine the lack of difference in carbonation conversion for CaO that was calcined at 1000 ℃ and 950 ℃” has been delete.
Page 9, lines 272-273.
Reviewer 2 Report (New Reviewer)
Review: Effect of steam on Carbonation of CaO in Ca-looping
This manuscript deals with the effect of steam on carbonation and calcination, separately and combined in a calcium looping system. An in-house built thermogravimetric analyser with high capacity was employed to assess the effect of steam concentration on both reactions. The manuscript could be of interest for the readership of this journal. However, major changes should be made to the original manuscript in order to make it fit for publication.
General comments:
1. Language should be thoroughly checked in order to avoid typos and grammar mistakes. Moreover, the figures should be double checked as there are some Figures where the symbols don’t match the colour of the line.
2. Literature is missing from both introduction and discussion. Calcium looping has been widely investigated by many research groups at different scales (from lab to large pilot). Steam addition has also been investigated by different research groups.
3. Claims are not supported by relevant literature in the discussion section. The discussion is lacking in depth, trends are not properly supported and reaction conditions are not explained (i.e. calcination is longer than carbonation when carbonation is a much slower reaction). All of the claims require further explanation and, as I stated before, need to be supported by relevant literature on the field.
The document should be proofread extensively
Author Response
1. Language should be thoroughly checked in order to avoid typos and grammar mistakes. Moreover, the figures should be double checked as there are some Figures where the symbols don’t match the colour of the line.
Response:
We are very sorry that there are many typos and grammar mistakes as well as picture errors in the manuscript. We will check the whole manuscript carefully and polish it for native English speakers.
2. Literature is missing from both introduction and discussion. Calcium looping has been widely investigated by many research groups at different scales (from lab to large pilot). Steam addition has also been investigated by different research groups.
Response:
Thanks to the reviewer for pointing out the problems in this manuscript, we have added the reference of other scholars' research results to the revised manuscript.
Page 2,lines 41-88; Page 5,line 162 and page 8,line 246.
References [2].[3].[6].[8].[9].[13].[14].[18].[19].[20].[21].[22].[30]
[2] Norbarzad MJ, Tahmasebpoor M, Heidari M, Pevida C. Theoretical and experimental study on the fluidity performance of hard-to-fluidize carbon nanotubes-based CO2 capture sorbents. Frontiers of Chemical Science and Engineering, 2022, 16(10): 1460-1475.
[3] Troya JA, Jiménez PSE, Perejón A, Moreno V, Valverde JM, Maqueda PAP. Kinetics and cyclability of limestone (CaCO3) in presence of steam during calcination in the CaL scheme for thermochemical energy storage. Chemical Engineering Journal, 2021, 417: 129194.
[6] Imani M, TPeridas G, Mordick Schmidt B. Improvement in cyclic CO2 capture performance and fluidization behavior of eggshell-derived CaCO3 particles modified with acetic acid used in calcium looping process. Journal of CO2 Utilization, 2022, 65: 102207.
[8] Heidari M, Tahmasebpoor M, Mousavi SB, Pevida C. CO2 capture activity of a novel CaO adsorbent stabilized with (ZrO2+Al2O3+CeO2)-based additive under mild and realistic calcium looping conditions.Journal of CO2 Utilization, 2021, 53: 101747.
[9] Mousavi SB, Heidari M, Rahmani F, Sene RA, Clough PT, Ozmen S. Highly robust ZrO2-stabilized CaO nanoadsorbent prepared via a facile one-pot MWCNT-template method for CO2 capture under realistic calcium looping conditions. Journal of Cleaner Production, 2023, 384: 135579.
[13] Heidari M, Tahmasebpoor M, Antzaras A, Lemonidou AA. CO2 capture and flfluidity performance of CaO-based sorbents: Effect of Zr, Al and Ce additives in tri-, bi- and mono-metallic confifigurations. Process Safety and Environmental Protection, 2020, 14: 349-365.
[14] Heidari M, Mousavi SB, Rahmani F, Clough PT, Ozmen S. The novel Carbon Nanotube-assisted development of highly porous CaZrO3-CaO xerogel with boosted sorption activity towards high-temperature cyclic CO2 capture. Energy Conversion and Management , 2022, 274: 116461.
3. Claims are not supported by relevant literature in the discussion section. The discussion is lacking in depth, trends are not properly supported and reaction conditions are not explained (i.e. calcination is longer than carbonation when carbonation is a much slower reaction). All of the claims require further explanation and, as I stated before, need to be supported by relevant literature on the field.
Response:
Thanks to the reviewer's valuable advice, we modified each figures and added experimental conditions.
According to the kinetic principle, the calcination rate is generally faster than the carbonation rate. However, it should be pointed out that carbonation is divided into a fast chemical reaction control stage and a slow diffusion control stage[6][14], and the fast chemical reaction stage contributes a large carbonation conversion rate[18]. For industrial application, The slow diffusion control phase makes little sense. Therefore, the control reaction time of the carbonation stage in this paper is 5min
Because a new self-made thermogravimetric analyzer was adopted in this manuscript to realize the calcium cycle experiment under the condition of rapid changes of temperature and reaction atmosphere switching, there are few relevant references.
References:
[6] Imani M, TPeridas G, Mordick Schmidt B. Improvement in cyclic CO2 capture performance and fluidization behavior of eggshell-derived CaCO3 particles modified with acetic acid used in calcium looping process. Journal of CO2 Utilization, 2022, 65: 102207.
[14] Heidari M, Mousavi SB, Rahmani F, Clough PT, Ozmen S. The novel Carbon Nanotube-assisted development of highly porous CaZrO3-CaO xerogel with boosted sorption activity towards high-temperature cyclic CO2 capture. Energy Conversion and Management, 2022, 274: 116461.
[18] Wang NN, Feng YC and Guo X. Atomistic mechanisms study of the carbonation reaction of CaO for high temperature CO2 capture. Applied Surface Science, 2020, 532: 147425.
Round 2
Reviewer 1 Report (Previous Reviewer 3)
This study is now ready to be published.
This manuscript is a resubmission of an earlier submission. The following is a list of the peer review reports and author responses from that submission.
Round 1
Reviewer 1 Report
The current work studied the influence of water vapor on the Ca-looping, calcination and carbonation process, and the self-made equipment that can realize rapid rise and drop temperature is adopted, which is closer to the practical application. It can be accepted for publication after addressed the following issues.
1. There should be more references about energy and chemical journals, such as MOLECULES.
2. The experimental system shown in Fig. 1 should be appropriately landscaped.
3. The reasons for calcination temperature and parameter selection should be explained in the introduction.
4. The writing of the full text and the format of the references should be carefully checked.
Author Response
Question1: There should be more references about energy and chemical journals, such as MOLECULES.
Response:
Thank you for kind reminder. This paper has updated the references and added the literature of energy journals. For example, References 3,5,11,13,17,27.
[3] Madejski P, Chmiel K, Subramanian N, Kus T. Methodsand Techniques for CO2 Capture:Review of Potential Solutions and Applications in Modern Energy Technologies. Energies.2022, 15, 887.[Google Scholar]
[5] Han R, Wang Y, Xing S, Pang C, Hao Y, Song C, Liu, Q. Progress in reducing calcination reaction temperature of Calcium-Looping CO2 capture technology: A critical review. Chem. Eng. J. 2022, 450, 137952. [Google Scholar] [CrossRef].
[11] Moreno J, Hornberger M, Schmid M, Scheffknecht G. Oxy-Fuel Combustion of Hard Coal, Wheat Straw and Solid Recovered Fuel in a 200 kWth Calcium Looping CFB Calciner. Energies,2021,14(8): 2162. [Google Scholar]
[17] Labus K. Comparison of the Properties of Natural Sorbents for the Calcium Looping Process. Materials 2021, 14(3), 548. [Google Scholar] [CrossRef]
[27] Li D, Wang Y, Li ZS. Limestone Calcination Kinetics in Microfluidized Bed Thermogravimetric Analysis (MFB-TGA) for Calcium Looping. Catalysts,2022, 12(12), 1661. [Google Scholar] [CrossRef]
Question2: The experimental system shown in Fig. 1 should be appropriately landscaped.
Response :
Thanks to the reviewers for their professional advice. This paper has optimized the experimental system in Figure1. And replace Figure 1 in the original paper.
Figure 1. The tube furnace experimental system.
Question3: The reasons for calcination temperature and parameter selection should be explained in the introduction.
Response:
Thanks for the valueable comments. In this paper, the calcination temperature and parameters have been added in the introduction. And mark it in blue in the original text, as follows, ‘Carbon dioxide in flue gas is absorbed by calcium oxide in a carbonizer at about 650 ℃. Part of the carbonated solid is transferred to the regeneration reactor (calciner). In the calcinator, the carbon dioxide bound to the solid phase is released at about 900 ℃, and the adsorbent is regenerated at the same time’.
Question4: The writing of the full text and the format of the references should be carefully checked.
Response:
Thanks to the reviewers for their professional advice. This paper has checked the full text and modified. In terms of grammar, the author has seriously revised it, and professional institutions have polished it with proof of polishing.The references and misquotes that are not referenced to the manuscript have been modified. And mark it in blue in the original text. For, example, line 144 on page 4 of the original,‘because at 1200 mL/min, the mass transfer is not the limiting factor of the reaction. References 2, 11, 16, 17, and 18 have been included in the paper’.

Reviewer 2 Report
The research lacks novelty and originality. English usage is poor and no novel results are reported. Furthermore, references to recent research on the same topic have been omitted. For these reasons, I do not consider this work deserves its publication in molecules.
Author Response
Reviewer #2 : The research lacks novelty and originality. English usage is poor and no novel results are reported. Furthermore, references to recent research on the same topic have been omitted. For these reasons, I do not consider this work deserves its publication in molecules.
Response:
Thanks for the reviewer 's suggestion. There are many studies on calcium-based chemical looping capture of CO2. However, under the condition of rapid heating and cooling, considering the influence of water vapor atmosphere is rarely reported. In terms of grammar, the author has seriously revised it, and professional institutions have polished it with proof of polishing.
Reviewer 3 Report
After reviewing this paper, I think it can be considered to publish if the following issues are solved:
Specific comments:
1- References 2, 11, 16, 17, and 18 have not been included in the paper. please edit the reference type according to the journal guideline.
2- The novelty section should be considerably boosted and revised.
3- Please add the drawbacks of using TGA in the introduction section (for example the fluidity of materials cannot be addressed through TGA)
4- The following sentence should be revised due to its grammatical errors:
“This flow rate was selected as preliminary 100 tests had indicated that at 1200 mL/min the mass transfer was not the limiting factor for 101 the reactions.”
In addition, please double-check the whole paper and edit all grammatical and typing errors: for example, in line 118, two disparate citation formats were used, and ref 21 is mentioned twice.
5- Researchers can evaluate the effect of steam on both carbonation stages, including kinetic and diffusion, separately, provide the practical data
6- Most references are related to the years before 2018, whereas numerous remarkable papers about CO2 capture through calcium looping condition has been published that surely improve the study. Accordingly, old-time refs can be replaced with up-to-date articles, and the relevant studies should be cited: 10.1016/j.jcou.2021.101747, 10.1016/j.psep.2020.07.041, 10.1016/j.enconman.2022.116461, 10.1016/j.jclepro.2022.135579, 10.1016/j.seppur.2022.122625, 10.1016/j.seppur.2022.122523, 10.1016/j.jcou.2022.102207, 10.1016/j.jcou.2022.102007.
7- The SEM micrographs of the spent sample after each condition are required to conduct visual assessments of the morphological sintering during the process. Please add the SEM analysis.
8- The main reason for improving the CaO conversion in the presence of H2O should be provided in the section related to Fig. 6.
9- The influence of water on the carbonation of limestone should be discussed more using previously published studies that investigated the effect of steams on the cyclic activity of CaO-based sorbents. From this point of view, the discussion section can be added as the last section of the Result section.
Author Response
Reviewer #3: After reviewing this paper, I think it can be considered to publish if the following issues are solved:
Response: Thank reviewers for their recognition of this article.
Question1: References 2, 11, 16, 17, and 18 have not been included in the paper. please edit the reference type according to the journal guideline.
Response:
Thanks to the reviewers for their professional advice. References have been modified according to the format of references according to the requirements of journals, and references 2,11,16,17 to the original text.
[2] Johnsson F, Kjärstad J, Rootzén, J. The threat to climate change mitigation posed by the abundance of fossil fuels. Clim. Policy 2019, 19, 258-274. [Google Scholar] [CrossRef]
[11] Moreno J, Hornberger M, Schmid M, Scheffknecht G. Oxy-Fuel Combustion of Hard Coal, Wheat Straw and Solid Recovered Fuel in a 200 kWth Calcium Looping CFB Calciner. Energies,2021,14(8): 2162. [Google Scholar]
[16] Tian SC, Yan, F, Zhang, ZT, Jiang JG. Calcium-looping reforming of methane realizes in situ CO2 utilization with improved energy efficiency. Science Advances. 2019,5(4): eaav5077. [CrossRef]
[17] Labus K. Comparison of the Properties of Natural Sorbents for the Calcium Looping Process. Materials 2021, 14(3), 548. [Google Scholar] [CrossRef]
Question2: The novelty section should be considerably boosted and revised.
Response:
Thanks to the reviewers for their professional advice. The highlight part 2 was modified to “study the effect of water vapor on the calcium cycle in detail’. And deleted the highlight part 3---The effect of steam on calcium cycle was studied from reaction temperature and mechanism..
Question3: Please add the drawbacks of using TGA in the introduction section (for example the fluidity of materials cannot be addressed through TGA)
Response :
Thanks to the reviewers for their professional advice. This article adds the disadvantages of using TGA in line 114 on page 3 of the introduction. And mark it in blue in the original text, as follows, “however, how the TGA method eliminates the mass transfer effect and maintains isothermal decomposition, especially at high temperatures, remains a challenge. In addition, the heating rate of the particles in the thermogravimetric analyzer is usually lower than 100K/min, and the diffusion resistance is serious, which is not conducive to the establishment of the kinetic model”.
Question4: The following sentence should be revised due to its grammatical errors: “This flow rate was selected as preliminary 100 tests had indicated that at 1200 mL/min the mass transfer was not the limiting factor for 101 the reactions.” In addition, please double-check the whole paper and edit all grammatical and typing errors: for example, in line 118, two disparate citation formats were used, and ref 21 is mentioned twice.
Response :
Thanks for the valueable comments. This article has modified the wrong syntax in the sentence to”because at 1200 mL/min, the mass transfer is not the limiting factor of the reaction.”And The grammar and words of the full text are checked.
Question5: Researchers can evaluate the effect of steam on both carbonation stages, including kinetic and diffusion, separately, provide the practical data
Response :
Thanks for the reviewer 's suggestion. Kinetic parameters are very important for calcium absorbents. However, this paper mainly examines the influence of water vapor. The effects of water vapor in the calcination stage, water vapor in the carbonation stage and water vapor at different reaction temperatures are considered respectively. The pore structure, and SEM images were tested, and the diffusion coefficient has been calculated on page 12 in section 3.4. There has been more content. In the future, we will study the effect of water vapor on Ca-based absorbent in more detail.
Question6: Most references are related to the years before 2018, whereas numerous remarkable papers about CO2 capture through calcium looping condition has been published that surely improve the study. Accordingly, old-time refs can be replaced with up-to-date articles, and the relevant studies should be cited: 10.1016/j.jcou.2021.101747,10.1016/j.psep.2020.07.041,10.1016/j.enconman.2022.116461,10.1016/j.jclepro.2022.135579,10.1016/j.seppur.2022.122625,10.1016/j.seppur.2022.122523, 10.1016/j.jcou.2022.102207, 10.1016/j.jcou.2022.102007.
Response :
Thank you for kind reminder. References in this manuscript have been updated, as follows, references2,3,5,6,8,11,13,14,15,16,17,27,30,25,26. Other references in the original text.
- Johnsson F, Kjärstad J, Rootzén, J. The threat to climate change mitigation posed by the abundance of fossil fuels. Clim. Policy 2019, 19, 258-274. [Google Scholar] [CrossRef]
- Madejski P, Chmiel K, Subramanian N, Kus T. Methodsand Techniques for CO2 Capture:Review of Potential Solutions and Applications in Modern Energy Technologies. Energies.2022, 15, 887.[Google Scholar]
[5] Han R, Wang Y, Xing S, Pang C, Hao Y, Song C, Liu, Q. Progress in reducing calcination reaction temperature of Calcium-Looping CO2 capture technology: A critical review. Chem. Eng. J. 2022, 450, 137952. [Google Scholar] [CrossRef].
[6] Chen J, Duan L, Sun S. Review on the development of sorbents for calcium looping. Energy&Fuels, 2020,34:7806–7836. [CrossRef]
[8] Peridas G, Mordick Schmidt B. The role of carbon capture and storage in the race to carbon neutrality. Electr. J. 2021, 34, 106996. [Google Scholar] [CrossRef]
Question7: The SEM micrographs of the spent sample after each condition are required to conduct visual assessments of the morphological sintering during the process. Please add the SEM analysis.
Response:
Thanks for the reviewer 's suggestion. In this paper, SEM data are supplemented in 352 lines of 12 pages. The details are as follows:
CaCO3 is dense and non-porous, showing a rich pore structure after calcination. After calcination in water vapor atmosphere, the particle size of CaCO3 increases and some small particles grow up. After carbonization in steam atmosphere, the surface of CaO is more compact than that of calcium carbonate, which corresponds to higher carbonation conversion.
Figure 12. The SEM images of samples under different conditions.(a)CaCO3; (b)CaO ; (c)CaO (calcined with water); (d)CaCO3 (calcined in water).
Question8: The main reason for improving the CaO conversion in the presence of H2O should be provided in the section related to Fig. 6.
Response :
Thanks to the reviewers for their professional advice. This paper supplements Fig.6 by adding the figure of calcium oxide conversion rate on page 8, line 257 of the original text.
Question 9: The influence of water on the carbonation of limestone should be discussed more using previously published studies that investigated the effect of steams on the cyclic activity of CaO-based sorbents. From this point of view, the discussion section can be added as the last section of the Result section.
Response :
Thanks for the reviewer 's suggestion.Starting from calcium carbonate, this paper first discusses the influence of water vapor in the calcination-carbonation stage, and then discusses the influence of water vapor in the calcination stage, carbonation stage and temperature change. If the influence of water vapor in the carbonation stage is added to the discussion section, the structure of the article will be destroyed. In the conclusion part, we add a more detailed discussion of the influence of water vapor in combination with previous studies.

Round 2
Reviewer 2 Report
In the first review report, I said that this work lacks novelty. The conclusions are already known and can be found in other works from the literature. I would not accept it.